# ⚖ Libra: Assessing and Improving Reward Model by Learning to Think

## Abstract

Reinforcement learning (RL) has significantly improved the reasoning ability of large language models. However, current reward models underperform in challenging reasoning scenarios and predominant RL training paradigms rely on rule-based or reference-based rewards, which impose two critical limitations: 1) the dependence on finely annotated reference answer to attain rewards; and 2) the requirement for constrained output format. These limitations fundamentally hinder further RL data scaling and sustained enhancement of model reasoning performance. To address these limitations, we propose a comprehensive framework for evaluating and improving the performance of reward models in complex reasoning scenarios. Through the proposed framework, we reveal the intrinsic strengths of generative reward models in reasoning tasks, highlighting their potential to excel in downstream scenarios involving unverifiable reasoning tasks. Specifically, we first present a reasoning-oriented benchmark (Libra Bench), systematically constructed from a diverse collection of challenging mathematical problems and advanced reasoning models, to address the limitations of existing reward model benchmarks in reasoning scenarios. We further introduce a novel approach for improving the generative reward model via learning-to-think methodologies, resulting in Libra-RM series, a collection of generative reward models that achieve state-of-the-art results on various benchmarks. Comprehensive downstream experiments are conducted and the experimental results demonstrate the correlation between our Libra Bench and downstream application, and the potential of Libra-RM to further improve reasoning models with unlabeled data.

## 1 Introduction

Recent advances in reinforcement learning (RL) and inference-time scaling have significantly unlocked the potential of large language models (LLMs), greatly enhancing their reasoning capabilities (OpenAI, 2024; Guo et al., 2025a). Unlike reinforcement learning from human feedback (RLHF) (Ouyang et al., 2022; Bai et al., 2022; Zheng et al., 2023b; Xiong et al., 2023), the current RL training paradigms for reasoning models predominantly rely on reference-based reward (Guo et al., 2025a; Yang et al., 2025; Seed et al., 2025; Lambert et al., 2024a). Despite the high accuracy, these methods rely on a finely annotated reference answer to attain rewards and a constrained output format to extract the key answer, which limits the use of large-scale data for general reinforcement learning.

To overcome these limitations, there is an urgent need to re-evaluate and advance the role of Reward Models (RMs) as robust proxies for human judgment, especially for general, unlabeled, or hard-to-standardize data. However, existing RMs and their associated benchmarks fall short in complex reasoning scenarios due to three key aspects: 1) existing RM benchmarks are insufficient to assess reward models in complex reasoning scenarios, due to the absence of challenging questions and responses from advanced reasoning models (Lambert et al., 2024b; Frick et al., 2024; Zhou et al., 2024; Tan et al., 2024; Zheng et al., 2024; Song et al., 2025; Liu et al., 2024b); 2) current RMs are designed without deep thinking capabilities and exhibit limited effectiveness when dealing with complex problems (Tan et al., 2024; Zheng et al., 2024; Liu et al., 2024c); 3) The traditional pairwise comparison learning objective of RMs often does not inherently align with the precise correctness metrics crucial for evaluating reasoning tasks (Yang et al., 2024; Liu et al., 2024c).

To address these limitations, we propose a comprehensive framework for evaluating and improving the performance of reward models in challenging reasoning scenarios. We first present a reasoning-oriented RM benchmark, namely Libra Bench, to alleviate the shortcomings of existing RM benchmarks. The Libra Bench is curated from a diverse collection of challenging mathematical problems and advanced reasoning models, and aims to assess pointwise judging accuracy in terms of correctness. These characteristics collectively ensure that our Libra Bench is well-aligned with the current research and development trajectory of reasoning models. Through our Libra Bench, we clearly observe and analyze the limitations of existing RMs in challenging reasoning scenarios.

Building upon these observations, we further introduce a novel approach to improving generative reward models through "learning-to-think" methodologies. This approach is founded on two core insights: 1) Long-CoT reasoning, i.e., inference-time scaling, has the potential to improve the accuracy of RM, especially in reasoning scenarios. 2) Taking the judging process as a verifiable task, we can further optimize generative reward models using techniques analogous to those applied to LLMs, such as rejection sampling and reinforcement learning.

Based on the proposed framework, we develop Libra-RM series, including Libra-RM-32B and Libra-RM-32B-MATH, a collection of generative reward models with deep thinking abilities. Extensive results demonstrate that our Libra-RM series achieves state-of-the-art performance on various RM benchmarks, especially on reasoning-oriented benchmarks such as Libra Bench. We further conduct comprehensive RL experiments to analyze our Libra Bench and Libra-RM. The experimental results demonstrate the correlation between our Libra Bench and downstream application, and the potential of Libra-RM in further reasoning data scaling with unlabeled data.

To summarize, our main contributions are as follows:

- We curate a reasoning-oriented RM benchmark from a diverse collection of challenging mathematical problems and advanced reasoning models, named Libra Bench, to address the limitations of existing RM benchmarks in reasoning scenarios.

- We propose a novel "learning-to-think" approach to enhancing generative reward models, yielding the powerful Libra-RM series, which achieves state-of-the-art results on multiple RM benchmarks.

- Our comprehensive RL experiments demonstrate a significant correlation between performance on Libra Bench and practical downstream applications, highlighting the potential of Libra-RM for enabling effective RL data scaling with unlabeled data.

## 2 RELATED WORK

**Reward Models**   Reward models (RMs) are designed to assign reward scores to responses generated by LLMs, and have been widely adopted in reinforcement learning, data selection, model evaluation, and other applications (Zheng et al., 2023b; Dong et al., 2024b; Zheng et al., 2023a; Li et al., 2024; Dubois et al., 2024; Gu et al., 2024; Seed et al., 2025). RMs are predominantly categorized into discriminative and generative types. Discriminative reward models typically consist of an LLM backbone coupled with a value head. They are trained on preference data with a classification objective and assign scalar rewards to responses (Liu et al., 2024a; Adler et al., 2024; Wang et al., 2024b). In contrast, generative reward models share the same architecture as standard LLMs but output textual judgments containing reward information for input responses (Zhang et al., 2024; Wang et al., 2024c; Zhu et al., 2023; Ankner et al., 2024; Liu et al., 2025). Notably, several works have proposed enhancing generative reward models with deep thinking capacities (Chen et al., 2025a;b; Whitehouse et al., 2025; Guo et al., 2025b). However, fully leveraging the advantages of inference-time scaling for reasoning tasks and realizing the potential of thinking-enhanced generative reward models in downstream applications remain significant challenges.

**Reward Model Benchmarks**   Reward model benchmarks play a crucial role in guiding RM optimization and forecasting their performance on downstream applications (Malik et al., 2025; Frick et al., 2024). Conventional RM benchmarks predominantly target general question-answering tasks, assessing a model's ability to select the superior response in a pairwise setting (Lambert et al., 2024b; Frick et al., 2024; Zhou et al., 2024; Tan et al., 2024; Liu et al., 2024b; Saha et al., 2025), which aligns with the Bradley-Terry (BT) model commonly employed in RM training (Bradley & Terry, 1952). This pairwise accuracy evaluation paradigm has been extended to other specific domains,

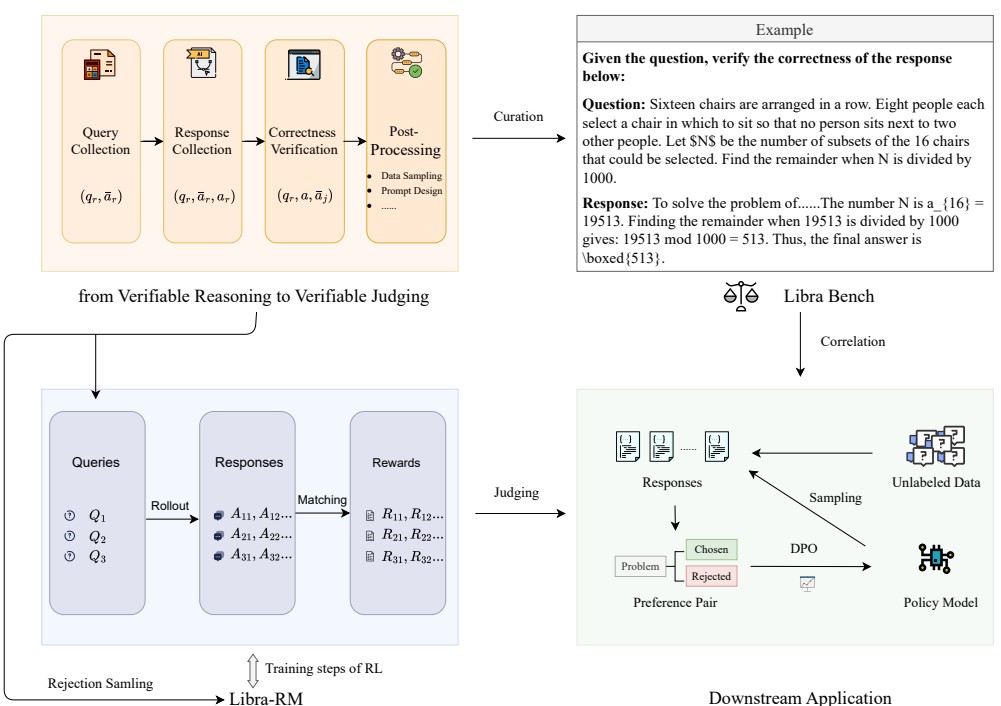

Figure 1: The overview of building Libra Bench and Libra-RM . For Libra Bench , we design the data strategy *from Verifiable Reasoning to Verifiable Judging*, to curate RM benchmark from a collection of challenging mathematical problems and advanced reasoning models. For Libra-RM, we adopt the same data strategy and combine reinforcement learning (RL) and rejection sampling for training.

including multimodal contexts, multilingual tasks, agentic systems, and more (Gureja et al., 2024; Lù et al., 2025; Jin et al., 2024; Wu et al., 2025; Chen et al., 2024; Yasunaga et al., 2025; Li et al., 2025; Ruan et al., 2025). Recently, reasoning-oriented RM benchmarks have been proposed to evaluate the accuracies of (process) reward models in reasoning tasks (Liu et al., 2024c; Zheng et al., 2024; Song et al., 2025). However, these existing RM benchmarks suffer from one or two major limitations: the absence of challenging questions and responses from advanced reasoning models, rendering them insufficient for assessing reward models in reasoning scenarios.

**Reinforcement Learning for LLMs**   Reinforcement learning (RL) is widely employed in the post-training stage to enhance reasoning capabilities and align models with human preferences (Ouyang et al., 2022; Guo et al., 2025a). Algorithms such as PPO, GRPO and their variants are predominantly used in RL for LLMs (Schulman et al., 2017; Shao et al., 2024; Yu et al., 2025; Yuan et al., 2025), while offline methods like DPO and KTO have also been proposed to accommodate resource-constrained environments (Rafailov et al., 2023; Ethayarajh et al., 2024). RL for LLMs can be further classified by reward source into Reinforcement Learning from Human Feedback (RLHF) and Reinforcement Learning from Verifiable Reward (RLVR). RLHF leverages a reward model trained on human preference data to provide reward signals (Ouyang et al., 2022; Bai et al., 2022; Zheng et al., 2023b; Xiong et al., 2023), while RLVR optimizes models on verifiable tasks and receives rewards from rule-based answer matching and other predefined scripts (Dong et al., 2024a; Lambert et al., 2024a; Guo et al., 2025a).

## 3   LIBRA BENCH

In this section, we detail the curation pipeline for Libra Bench (Figure 1) and present its primary statistics and analysis. Distinct from existing RM benchmarks, our Libra Bench is constructed from a diverse set of challenging mathematical problems and advanced reasoning models, and is designed to

assess pointwise accuracy in terms of correctness. These attributes ensure that Libra Bench is well aligned with contemporary research, where reasoning models are primarily assessed and optimized for correctness on complex reasoning tasks.

### 3.1 Pipeline: from Verifiable Reasoning to Verifiable Judging

As illustrated in Figure 1, we curate the Libra Bench with the strategy: from **V**erifiable reasoning to **V**erifiable judging (V2V), for the RM evaluation. The total curation process consists of four stages: query collection, response collection, correctness verification and post-processing.

**Query Collection**   The query collection serves as the starting point of the entire curation pipeline. To adapt to the development of reasoning models, we collect 204 challenging mathematical problems from MATH-500 level5 (Lightman et al. (2023)), AIME 2024, and AIME 2025. Each problem is paired with a golden reference answer, covering various formats including integers, fractions, and formulas. Formally, each verifiable reasoning instance is denoted as $(q_r, \bar{a}_r)$, where $q_r$ is the reasoning problem and $\bar{a}_r$ is its golden reference answer.

**Response Collection**   Compared with existing RM benchmarks, we rollout generations from a collection of advanced reasoning models to assess the capacity of RM in complex reasoning tasks, including DeepSeek-R1 (Guo et al., 2025a), Qwen3-32B (Yang et al., 2025), QwQ-32B (Qwen, 2025), DeepSeek-R1-Distill-Qwen-7B and DeepSeek-R1-Distill-Qwen-1.5B (Guo et al., 2025a). These models exhibit a wide range of accuracies (28.9% - 81.4% on AIME 2024), ensuring the diversity of our Libra Bench. For each problem $q_r$, we sample at least 64 responses from each model to guarantee a sufficient number of both correct and incorrect replies. At this stage, each data point is formulated as $(q_r, \bar{a}_r, a_r)$, where $a_r$ is the sampled response for $q_r$.

**Correctness Verification**   We annotate the outcome correctness of each response $a_r$ based on the problem $q_r$ and the reference answer $\bar{a}_r$, thereby transforming reasoning problems to judging problems. In practice, we employ a sophisticated combination of different methods to ensure the reliability of correctness verification, including rule-based answer matching, model-based evaluation and human annotation (see Appendix C.1 for details). We denote the label of correctness as $\bar{a}_j$ which takes binary values 0 or 1. Each sample is thus represented as $(q_r, a_r, \bar{a}_j)$, where the $\bar{a}_r$ is omitted after annotation as in existing RM benchmarks. The change of subscript from $r$ to $j$ indicates a transition from **r**easoning problem to a **j**udging problem.

**Post-processing**   We further perform several post-processing steps to refine our Libra Bench. First, we remove the Chain-of-Thought (CoT) segments from sampled responses, as they often involve complex trial-and-error processes that are not supervised in mainstream training paradigms (Guo et al., 2025a). The truncated samples containing only the CoT component are also filtered out. Secondly, we balance the proportion of our Libra Bench such that each model contributes an equal number of correct and incorrect responses in each data subset, as detailed in Table 1. For evaluation, the RM receives the concatenation of $q_r$, $a_r$, and the predefined prompt template as input, determines the correctness of $a_r$, and outputs a binary prediction. Benefiting from the balanced distribution, we directly calculate the accuracies across different data subsets to assess the capacity of RM in reasoning scenarios.

### 3.2 Statistics and Analysis

We present the basic statistics of our Libra Bench in Table 1. Our Libra Bench consists of 3,740 samples which are curated from 204 challenging mathematical problems and 5 advanced reasoning models. More examples of our Libra Bench can be found in Appendix C.2.

We further evaluate state-of-the-art reward models and LLM-as-a-Judge methods in our Libra Bench, and report their performance in Table 2. Compared with the reasoning subsets of existing RM benchmarks, most models achieve lower accuracy on our Libra Bench, owing to both the increased difficulty of the problems and the presence of confusing responses from advanced reasoning models. From Table 2, we can also observe the superior performance of thinking models over non-thinking models on our Libra Bench. These findings motivate further improvements in RM accuracy via learning-to-think methodologies, which will be discussed in Section 4.

| | MATH | | AIME 2024 | | AIME 2025 | |
|---|---|---|---|---|---|---|
| | **correct** | **incorrect** | **correct** | **incorrect** | **correct** | **incorrect** |
| **Number of Problems** | 134 | 134 | 30 | 30 | 30 | 30 |
| **Number of Samples** | 680 | 680 | 600 | 600 | 600 | 600 |
| DeepSeek-R1 | 134 | 134 | 120 | 120 | 120 | 120 |
| Qwen3-32B | 134 | 134 | 120 | 120 | 120 | 120 |
| QwQ-32B | 134 | 134 | 120 | 120 | 120 | 120 |
| R1-Distill-Qwen-7B | 134 | 134 | 120 | 120 | 120 | 120 |
| R1-Distill-Qwen-1.5B | 134 | 134 | 120 | 120 | 120 | 120 |
| **Annotation Approach** | Model-based + Human | | Rule-Based | | Rule-Based | |

Table 1: Statistics of the Libra Bench. The Libra Bench comprises problems sourced from MATH-500 level5, AIME 2024, AIME 2025, with responses generated from Deepseek-R1, Qwen3-32B, QwQ-32B, DeepSeek-R1-Distill-Qwen-7B, and DeepSeek-R1-Distill-Qwen-1.5B. We divide the Libra Bench into three subsets based on the problem sources.

## 4 APPROACH FOR LIBRA-RM

To overcome the limitations of current RMs in reasoning scenarios, we propose a comprehensive approach to improve generative reward models via learning-to-think, resulting in our Libra-RM series. As illustrated in Figure 1, our Libra-RM series is trained through a combination of rejection sampling and reinforcement learning, sharing the same data strategy as our Libra Bench: from **V**erifiable reasoning to **V**erifiable Judging (V2V). The training setups are reported in Appendix E and ablation studies are discussed in Appendix D.

### 4.1 TASK DEFINITION

Conventional reward models (RMs) are typically discriminative, mapping an input query and a candidate response to a scalar quality score. In contrast, we explore a generative paradigm for reward modeling. A generative RM is a text-to-text model conditioned on a query $q$, a set of candidate responses $[a_1, a_2, \ldots]$, and a set of evaluation criteria $c$. Its objective is to generate a natural language judgment $j$ that evaluates the candidate answers according to the specified criteria. Formally, this process is defined as:

$$j = \mathrm{RM}_{\mathrm{gen}}(q, [a_1, a_2, \ldots], c) \tag{1}$$

Based on the criteria $c$, the judging tasks are further categorized into scoring and ranking. In the scoring setting, the generative reward model is required to assign a specific rating $score_i$ to each answer $a_i$. In contrast, the ranking task only requires the model to define the relative $rank_i$ for $a_i$ within the answer list. Both the scores and rankings can be extracted from the textual judgment $j$.

$$\mathrm{RM}(q, [a_1, \ldots], c) = \begin{cases} [score_1, \ldots] & c \in \mathbb{C}_{score} \\ [rank_1, \ldots] & c \in \mathbb{C}_{rank} \end{cases} \tag{2}$$

In this work, we develop Libra-RM-32B-MATH and Libra-RM-32B, both endowed with advanced deep thinking capabilities. The Libra-RM-32B-MATH is specialized for the reasoning-oriented pointwise scoring task in terms of correctness, achieving state-of-the-art performance on Libra Bench and downstream applications. The Libra-RM-32B is an extended version, which is also capable of preference ranking tasks as existing RMs, further demonstrating the generalizability of our approach.

### 4.2 REJECTION SAMPLING AND SUPERVISED FINE-TUNING

We perform rejection sampling and finetune our Libra-RM from the pretrained model Qwen2.5-32B to accelerate convergence and improve accuracy. During this phase, both judging and non-judging data are collected to enhance the diversity of the dataset and boost the performance of our Libra-RM.

**Judging Data**   For pointwise scoring, we initially collect a set of labeled verifiable judging data in reasoning scenarios via the V2V strategy. The correctness labels are annotated through model-based evaluation, employing advanced reasoning models to verify the answer correctness against the golden

reference answer. We then perform rejection sampling on DeepSeek-R1 to collect responses that are consistent with $\bar{a}_j$, forming the subset $\mathcal{D}_{score}^{rs}$. For pairwise ranking, we directly utilize existing preference data for RM training. The input prompt is formed by the concatenation of the problem $q$, answer pair $[a_1, a_2]$ and the judgment criteria $c_{rank}$. The golden references for judgment are taken from the original annotation in preference data. We apply the same rejection sampling procedure as in the pointwise scoring data to construct the data subset $\mathcal{D}_{rank}^{rs}$.

**Non-Judging Data** The curation of non-judging data adheres to the standard SFT setting, with prompts comprising only the question and ground truths as direct answers. For reasoning tasks, we perform rejection sampling on verifiable reasoning problems. And for general tasks (non-reasoning), we directly sample generations from DeepSeek-R1 as the ground truth without rejection, thereby ensuring the entire training process is RM-free. In this way, we obtain two data subsets $\mathcal{D}_{reason}^{rs}$ and $\mathcal{D}_{general}^{rs}$. In practice, we combine the $\mathcal{D}_{score}^{rs}$ and $\mathcal{D}_{reason}^{rs}$ subsets to train Libra-RM-32B-Math, while all data subsets are utilized for training Libra-RM-32B.

### 4.3 REINFORCEMENT LEARNING FOR JUDGING

Following rejection sampling and supervised fine-tuning, we further apply rule-based reinforcement learning on a verifiable dataset to improve the accuracy of our Libra-RM. We detail our training recipe from three aspects: data, reward design, and learning objective.

**Data** Similar to the rejection sampling, we curate a mixed RL dataset consisting of judging data and non-judging data for training. The judging data consists of $\mathcal{D}_{score}^{rl}$ for pointwise scoring and $\mathcal{D}_{rank}^{rl}$ for pairwise ranking, while the non-judging data consists solely of verifiable reasoning data $\mathcal{D}_{reason}^{rl}$. All of our RL dataset are verifiable, and the entire training process does not depend on any other RMs. In practice, we combine $\mathcal{D}_{score}^{rl}$ and $\mathcal{D}_{reason}^{rl}$ subsets for training Libra-RM-32B-MATH, while all three subsets are incorporated for training Libra-RM-32B.

**Reward Design** We adopt a rule-based reward signal that consists of correctness reward and length penalty for training Libra-RM, formulated as:

$$\text{reward}(x, y, \bar{y}) = \text{is\_correct}(y, \bar{y}) - \text{len\_penalty}(y) \tag{3}$$

where $x$ denotes the input, $y$ denotes the response, and $\bar{y}$ denotes the ground-truth answer for $x$. The correctness reward is\_correct$(y, \bar{y})$ is computed via rule-based answer matching to assess the correctness of the response's final outcome. If the extracted outcome aligns with the golden reference answer, a correctness reward of 1 is assigned; otherwise, a reward of 0 is given.

We also incorporate a length penalty into our reward system, which has been demonstrated to be effective in performance improvement and length compression (Yu et al., 2025; Team et al., 2025). As formulated in Equation 4, the length penalty is defined as the ratio of the excess length over the expected length to the buffer length, where $L_{exp}$ denotes the expected length and $L_{max}$ represents the maximum generated length during training. The buffer length is given by the difference $L_{max} - L_{exp}$, and the length penalty is constrained to the range $[0, 1]$ since responses longer than $L_{max}$ are truncated.

$$\text{len\_penalty}(y) = \max\left(\frac{|y| - L_{exp}}{L_{max} - L_{exp}}, 0\right) \tag{4}$$

**Learning Objective** We adopt GRPO (Shao et al., 2024) with the Clip-Higher strategy (Seed et al., 2025) as our reinforcement learning algorithm, motivated by its resource efficiency and strong empirical performance. The learning objective can be formulated as follows:

$$J_{\text{GRPO}}(\theta) = \mathbb{E}[q \sim P(Q), \{o_i\}_{i=1}^G \sim \pi_{\theta_{old}}(O|q)]$$

$$\frac{1}{G}\sum_{i=1}^G \frac{1}{|o_i|}\sum_{t=1}^{|o_i|}\left(\min\left(r_{i,t}(\theta)A_{i,t}, \text{clip}\left(r_{i,t}(\theta), 1-\varepsilon_{low}, 1+\varepsilon_{high}\right)A_{i,t}\right) - \beta D_{\text{KL}}\left(\pi_\theta||\pi_{ref}\right)\right)$$

$$\tag{5}$$

where $r_{i,t}(\theta) = \frac{\pi_\theta(o_{i,t}|q, o_{<t})}{\pi_{\theta_{old}}(o_{i,t}|q, o_{<t})}$, and $D_{\text{KL}}$ denotes an unbiased estimator of the KL divergence, formulated as $D_{\text{KL}}\left(\pi_\theta||\pi_{ref}\right) = \frac{\pi_{ref}(o_i|q)}{\pi_\theta(o_i|q)} - \log\frac{\pi_{ref}(o_i|q)}{\pi_\theta(o_i|q)} - 1$. The advantage $a_{i,t}$ represents the relative reward of the output within the corresponding group, calculated as $A_{i,t} = \frac{r_i - \text{mean}(\{r_1, r_2, \cdots, r_G\})}{\text{std}(\{r_1, r_2, \cdots, r_G\})}$.

| Model | MATH-500 | AIME2024 | AIME2025 | Average |
|---|---|---|---|---|
| *Discriminative Reward Models* | | | | |
| InternLM2-20B-Reward | 59.9 | 67.1 | 62.2 | 63.1 |
| Skywork-Reward-Gemma-2-27B | 55.8 | 54.5 | 55.1 | 55.1 |
| ArmoRM-8B-v0.1 | 57.2 | 61.8 | 58.9 | 59.3 |
| Qwen2.5-Math-RM-72B | 69.9 | 69.1 | 58.0 | 65.7 |
| AceMath-72B-RM | 73.6 | 65.4 | 60.8 | 66.6 |
| *LLM-as-a-Judge* | | | | |
| GPT-4o-0816 | 69.9 | 66.1 | 61.4 | 65.8 |
| GPT-4.1 | 71.3 | 71.0 | 65.0 | 69.1 |
| Claude-3.5-sonnet | 64.9 | 65.2 | 63.9 | 64.7 |
| Claude-3.7-sonnet | 70.8 | 65.6 | 65.0 | 67.1 |
| Llama-3.1-70B-Instruct | 50.8 | 50.4 | 51.4 | 50.9 |
| *LLM-as-a-Judge with thinking* | | | | |
| DeepSeek-R1 | 82.2 | 76.8 | 77.4 | 78.8 |
| Qwen3-32B | 80.2 | 78.3 | 75.8 | 78.1 |
| QwQ-32B | 80.8 | 77.1 | 74.7 | 77.5 |
| R1-Distill-Qwen-32B | 75.9 | 75.0 | 70.2 | 73.7 |
| *Generative Reward Models* | | | | |
| Skywork-Critic-Llama-3.1-70B | 55.4 | 60.6 | 57.2 | 57.7 |
| Libra-RM-32B-MATH (Ours) | **83.4** | **81.5** | **80.3** | **81.7** |
| Libra-RM-32B (Ours) | 82.8 | 79.7 | 77.5 | 80.0 |

Table 2: Evaluations on Libra Bench. Bold numbers indicate the best performance, while underlined numbers indicate the second-best performance among baseline models and our Libra-RM. For generative reward model, the best accuracy across different prompt templates is reported (see Appendix B). For discriminative reward model, we select a threshold that maximizes the average accuracy to convert model outputs into binary correctness predictions. Note that all metrics reported in the table are accuracy, and the subsets MATH-500, AIME2024, and AIME2025 only refer to the sources of problems.

## 5 EXPERIMENTS ON RM BENCHMARKS

We conduct extensive experiments to evaluate and analyze both our Libra Bench and our Libra-RM. We begin by detailing our experimental setups in subsection 5.1. Subsequently, we present the performance of Libra-RM and various baseline methods on Libra Bench in subsection 5.2. Furthermore, subsection 5.3 extends the evaluation to other widely adopted RM benchmarks, highlighting the generalizability of our approach.

### 5.1 EXPERIMENTAL SETUPS

**Benchmarks**   To ensure a comprehensive comparison, We evaluate our Libra-RM and baseline models on various RM benchmarks, including our Libra Bench(see section 3 for details), Reward Bench (Lambert et al., 2024b), PPE Preference, PPE Correctness (Frick et al., 2024), RMB (Zhou et al., 2024), and JudgeBench (Tan et al., 2024).

**Baseline methods**   We compare our Libra-RM with leading RMs and LLM-as-a-Judge methods:

- Discriminative Reward Models: InternLM2-20B-Reward  (Cai et al., 2024), Skywork-Reward-Gemma-2-27B (Liu et al., 2024a), ArmoRM-8B-v0.1 (Wang et al., 2024a), Nemotron-4-340B-Reward (Adler et al., 2024), Qwen2.5-Math-RM-72B (Yang et al., 2024), AceMath-72B-RM (Liu et al., 2024c).

- Generative Reward Models: Skywork-Critic-Llama-3.1-70B(Shiwen et al., 2024), DeepSeek-GRM-27B(Liu et al., 2025).

- LLM-as-a-Judge methods without thinking: GPT-4o-0816 (Hurst et al., 2024), GPT-4.1 (OpenAI, 2025), Claude-3.5-sonnet (Anthropic, 2024a), Claude-3.7-sonnet (Anthropic, 2024b), Gemini-1.5-pro (Team et al., 2024), Llama-3.1-70B-Instruct (Grattafiori et al., 2024).

| Model | Reward Bench | PPE-P | PPE-C | RMB | JudgeBench | Average |
|---|---|---|---|---|---|---|
| Discriminative Reward Model | | | | | | |
| InternLM2-20B-Reward | 90.2 | 61.0 | 63.0 | 62.9 | 63.4 | 68.1 |
| Skywork-Reward-Gemma-2-27B | **94.3** | 56.6 | 56.6 | 60.2 | 64.3 | 66.4 |
| ArmoRM-8B-v0.1 | 90.4 | 60.6 | 61.2 | 64.6 | 56.9 | 66.7 |
| Nemotron-4-340B-Reward | 92.0 | 59.3 | 60.8 | 69.9 | - | - |
| LLM-as-a-Judge | | | | | | |
| GPT-4o-0816 | 86.7 | **67.7** | - | - | 56.6 | - |
| Claude-3.5-sonnet | 84.2 | 67.3 | 68.4 | 70.6 | 64.3 | 71.0 |
| Gemini-1.5-pro | 86.8 | 66.1 | 59.8 | 56.5 | 47.1 | 63.3 |
| Llama-3.1-70B-Instruct | 84.0 | 65.3 | 63.2 | 68.9 | 52.3 | 66.7 |
| Generative Reward Model | | | | | | |
| DeepSeek-GRM-27B | 86.0 | 64.7 | 59.8 | 69.0 | - | - |
| Libra-RM-32B-MATH (Ours) | 89.1 | 63.9 | 75.2 | 65.5 | 76.6 | 74.1 |
| Libra-RM-32B (Ours) | 92.9 | 66.5 | **77.3** | **72.9** | **77.1** | **77.3** |

Table 3: Overall evaluations on mainstream reward model (RM) benchmarks for pairwise ranking tasks. PPE-P is short for PPE preference, and PPE-C is short for PPE correctness. Bold numbers indicate the best performance, while underlined numbers indicate the second-best performance. Baseline results are taken from previous work (Liu et al., 2025; Frick et al., 2024; Zhou et al., 2024), with missing JudgeBench scores supplemented by us.

- LLM-as-a-Judge methods with thinking: DeepSeek-R1 (Guo et al., 2025a), Qwen3-32B (Yang et al., 2025), QwQ-32B (Qwen, 2025), R1-Distill-Qwen-32B (Guo et al., 2025a).

- Libra-RM Series: Our proposed generative Reward Models with deep thinking capabilities. Libra-RM series was trained through the approach in 4, and training details are reported in E.

## 5.2 EXPERIMENTAL RESULTS ON LIBRA BENCH

We first compare the performance of our Libra-RM with various baseline models on the Libra Bench. As shown in Table 2, our Libra-RM-32B-MATH and Libra-RM-32B consistently outperform all baselines across all subsets of Libra Bench. Notably, Libra-RM-32B-MATH even surpasses Qwen3-32B and DeepSeek R1, which were trained from stronger base models, achieving accuracy gains of 3.6 and 2.9, respectively. To further understand these performance differences, we analyze the confusion matrices on Libra Bench, as shown in Table 8. The results indicate that verifying incorrect samples is substantially more challenging than verifying correct ones. Notably, our Libra-RM series demonstrates superior performance in handling incorrect samples and achieves the highest macro F1 score among all evaluated models.

## 5.3 EXPERIMENTAL RESULTS ON GENERAL RM BENCHMARKS

We further evaluate our Libra-RM with existing reward models and LLM-as-a-Judge methods on widely used RM benchmarks for comprehensive assessment, including Reward Bench (Lambert et al., 2024b), PPE Preference, PPE Correctness (Frick et al., 2024), RMB (Zhou et al., 2024), and JudgeBench (Tan et al., 2024).As shown in Table 3, our Libra-RM-32B outperforms both existing reward models and LLM-as-a-Judge methods in terms of average accuracy. Notably, compared with existing methods, Libra-RM-32B exhibits strong stability and delivers outstanding performance on all RM benchmarks. Table 4 further demonstrates the advantages of our Libra-RM on the reasoning subsets of these RM benchmarks. Detailed scores of our model and the baseline models are presented in Appendix A.

## 6 EXPERIMENTS ON DOWNSTREAM TASKS

We further conduct a series of DPO experiments in unverifiable reasoning settings, revealing the promising prospects of our Libra-RM and Libra Bench. Our experiments are based on R1-Distill-Qwen-7B and R1-Distill-Llama-8B as initial policy models, with queries drawn from Skywork-

| Model | AIME24 | AIME25 | RM ACC |
|---|---|---|---|
| R1-Distill-Qwen-7B† | 55.5 | 39.2 | - |
| + DPO×Skywork-RM | 54.8 | 39.8 | 55.1 |
| + DPO×Skywork-Critic | 54.2 | 41.6 | 57.7 |
| + DPO×GPT-4o-0816 | 54.6 | 42.3 | 65.8 |
| + DPO×AceMath-RM | 55.2 | 41.8 | 66.6 |
| + DPO×R1-Qwen-32B | 56.9 | 40.9 | 73.7 |
| + DPO×Qwen3-32B | 55.0 | 42.6 | 78.1 |
| + DPO×Libra-MATH | 57.7 | 43.3 | 81.7 |
| R1-Distill-Llama-8B‡ | 43.1(50.4) | 30.7 | - |
| + DPO×Skywork-RM | 41.9 | 30.0 | 55.1 |
| + DPO×Skywork-Critic | 45.5 | 29.6 | 57.7 |
| + DPO×GPT-4o-0816 | 47.5 | 31.6 | 65.8 |
| + DPO×AceMath-RM | 48.3 | 32.9 | 66.6 |
| + DPO×R1-Qwen-32B | 48.8 | 30.8 | 73.7 |
| + DPO×Qwen3-32B | 47.6 | 32.7 | 78.1 |
| + DPO×Libra-MATH | 48.5 | 35.4 | 81.7 |

Figure 2: Correlation between Libra Bench accuracy and downstream performance. †: Results taken from Guo et al. (2025a) and Wen et al. (2025). ‡: We re-evaluate the metrics for R1-Distill-Llama-8B, with the results in parentheses taken from Guo et al. (2025a). For brevity, we use the following abbreviations: "Skywork-RM" refers to Skywork-Reward-Gemma-2-27B, "Skywork-Critic" to Skywork-Critic-Llama-3.1-70B, "AceMath-RM" to AceMath-72B-RM, "R1-Qwen-32B" to R1-Distill-Qwen-32B, "Libra-MATH" to Libra-RM-32B-MATH, and "RM ACC" to RM accuracy on Libra Bench.

OR1-RL-Data (He et al., 2025) (see Appendix E.3 for details). As shown in Figure 2, our Libra-RM-32B-MATH achieves the best performance on the downstream DPO experiments, substantially improving the accuracy of both R1-Distill-Qwen-7B and R1-Distill-Llama-8B. Notably, all these enhancements are achieved without access to golden reference answers, demonstrating the potential of our Libra-RM for RL data scaling on unlabeled data. Furthermore, Figure 2 also reveals a clear correlation between Libra Bench accuracy and downstream application performance, highlighting the utility of our Libra Bench in guiding RM optimization and predicting RM performance.

Despite the crucial role of verifiable data in reasoning capabilities construction, our experiments yield several key insights for future researches: (1) As verifiable data becomes exhausted, reasoning and other scenarios increasingly rely on RMs to evaluate unverifiable synthetic data. Our experiments show that our Libra-RM-32B-MATH, initialized from a 32B base model, can effectively distinguish responses from leading reasoning models (see Table 2), and significantly enhance the performance of downstream applications. (2) Furthermore, we also observe that our Libra-RM-32B-MATH is able to identify errors in the reasoning process and assign differentiated scores based on the severity of these errors (see Appendix F for details). This finding offers new perspectives for further refining training methodologies and enhancing the overall user interaction experience.

# 7 CONCLUSIONS

In this paper, we present a comprehensive framework for evaluating and improving the performance of generative reward models in complex reasoning scenarios, introducing our Libra Bench and Libra-RM series. Distinct from existing RM benchmarks, the Libra Bench is curated from a diverse collection of challenging mathematical problems and advanced reasoning models, and aims to assess pointwise judging accuracy with respect to correctness. The Libra-RM series, including Libra-RM-32B and Libra-RM-32B-MATH, is trained through a combination of SFT and RL, where the judging process is formulated as a verifiable task. Systematic evaluations demonstrate that our Libra-RM series achieve state-of-the-art results on various benchmarks, especially in reasoning tasks. We also provide detailed ablation studies to further validate our approach. Furthermore, comprehensive downstream experimental results reveal the correlation between our Libra Bench and downstream application, as well as the potential of Libra-RM to further improve reasoning models with unlabeled data.

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

# A    DETAILED SCORES ON RM BENCHMARKS

In this section, we present further details regarding our evaluation results. Table 4 offers a comprehensive comparison of the performance of the Libra-RM and baseline models on the reasoning subset across Reward Bench, JudgeBench, and PPE correctness metrics. Owing to the deep thinking capacity and sophisticated training methodology, our Libra-RM series achieve state-of-the-art results on nearly every reasoning subset.

Detailed scores on RMB, PPE correctness and Reward Bench are illustrated on Table 5, Table 6, and Table 7. Table 8 reports the confusion matrices on Libra Bench.

| Model | Reward Bench Reasoning | PPE MATH | PPE GPQA | PPE MBPP | Judgebench |
|---|---|---|---|---|---|
| Discriminative Reward Model | | | | | |
| InternLM2-20B-Reward | 95.8 | 70.0 | 57.0 | 58.0 | 63.4 |
| Skywork-Reward-Gemma-2-27B | **98.1** | 63.0 | 53.0 | 59.0 | 64.3 |
| ArmoRM-8B-v0.1 | 97.3 | 71.0 | 57.0 | 54.0 | 56.9 |
| Nemotron-4-340B-Reward | 93.6 | 65.0 | 57.0 | 49.0 | |
| LLM-as-a-Judge & Generative Reward Model | | | | | |
| Claude-3.5-sonnet | 84.7 | 86.0 | 63.0 | 54.0 | 64.3 |
| Llama-3.1-70B-Instruct | 86.0 | 73.0 | 56.0 | 58.0 | 52.3 |
| DeepSeek-GRM-27B | 83.8 | 68.8 | 55.6 | 50.1 | |
| Libra-RM-32B-MATH (Ours) | 95.1 | 92.8 | 67.5 | **70.2** | 76.6 |
| Libra-RM-32B (Ours) | 97.2 | **96.3** | **71.1** | 67.4 | **77.1** |

Table 4: Detailed Scores on the reasoning subsets of existing RM benchmarks

We also present fine-grained evaluation results on Reward Bench, RMB, and PPE correctness, as summarized in Table 5, Table 6, and Table 7, respectively.

| Model | Helpful BoN | Helpful Pair | Harmful BoN | Harmful pair | Overall |
|---|---|---|---|---|---|
| Discriminative Reward Model | | | | | |
| InternLM2-20B-Reward | 58.5 | 76.3 | 49.9 | 67.0 | 62.9 |
| Skywork-Reward-Gemma-2-27B | 47.2 | 65.3 | 56.1 | 72.1 | 60.2 |
| ArmoRM-8B-v0.1 | 63.6 | 78.7 | 49.7 | 66.3 | 64.6 |
| LLM-as-a-Judge & Generative Reward Model | | | | | |
| Claude-3.5-sonnet | 70.5 | 83.8 | 51.8 | 76.4 | 70.6 |
| Gemini-1.5-pro | 53.6 | 76.3 | 29.9 | 66.1 | 56.5 |
| Llama-3.1-70B-Instruct | 64.8 | 81.1 | 55.8 | 73.9 | 68.9 |
| DeepSeek-GRM-27B | 62.3 | 80.5 | 57.0 | 76.1 | 69.0 |
| Libra-RM-32B-MATH (Ours) | 57.9 | 73.8 | 59.8 | 70.6 | 65.5 |
| Libra-RM-32B (Ours) | 64.8 | 79.6 | 67.5 | 79.5 | 72.9 |

Table 5: Detailed Scores on RMB

| Model | MMLU-Pro | MATH | GPQA | MBPP-Plus | IFEval | Mean |
|---|---|---|---|---|---|---|
| Discriminative Reward Model | | | | | | |
| InternLM2-20B-Reward | 68.0 | 70.0 | 57.0 | 58.0 | 62.0 | 63.0 |
| Skywork-Reward-Gemma-2-27B | 54.0 | 63.0 | 53.0 | 59.0 | 54.0 | 56.6 |
| ArmoRM-8B-v0.1 | 66.0 | 71.0 | 57.0 | 54.0 | 58.0 | 61.2 |
| Nemotron-4-340B-Reward | 70.0 | 65.0 | 57.0 | 49.0 | 63.0 | 60.8 |
| LLM-as-a-Judge & Generative Reward Model | | | | | | |
| Claude-3.5-sonnet | 81.0 | 86.0 | 63.0 | 54.0 | 58.0 | 68.4 |
| Llama-3.1-70B-Instruct | 73.0 | 73.0 | 56.0 | 58.0 | 56.0 | 63.2 |
| DeepSeek-GRM-27B | 64.8 | 68.8 | 55.6 | 50.1 | 59.8 | 59.8 |
| Libra-RM-32B-MATH (Ours) | 82.7 | 92.8 | 67.5 | 70.2 | 62.6 | 75.2 |
| Libra-RM-32B (Ours) | 86.1 | 96.3 | 71.1 | 67.4 | 65.6 | 77.3 |

Table 6: Detailed Scores on PPE Correctness

| Model | Chat | Chat Hard | Safe | Reason | Score |
|---|---|---|---|---|---|
| *Discriminative Reward Model* | | | | | |
| InternLM2-20B-Reward | 98.9 | 76.5 | 89.5 | 95.8 | 90.2 |
| Skywork-Reward-Gemma-2-27B | 96.1 | 89.9 | 93.0 | 98.1 | 94.3 |
| ArmoRM-8B-v0.1 | 96.9 | 76.8 | 90.5 | 97.3 | 90.4 |
| Nemotron-4-340B-Reward | 95.8 | 87.1 | 91.5 | 93.6 | 92.0 |
| *LLM-as-a-Judge & Generative Reward Model* | | | | | |
| GPT-4o-0816 | 96.1 | 76.1 | 88.1 | 86.6 | 86.7 |
| Claude-3.5-sonnet | 96.4 | 74.0 | 81.6 | 84.7 | 84.2 |
| Gemini-1.5-pro | 94.1 | 77.0 | 85.8 | 90.2 | 86.8 |
| Llama-3.1-70B-Instruct | 97.2 | 70.2 | 82.8 | 86.0 | 84.0 |
| DeepSeek-GRM-27B | 94.1 | 78.3 | 88.0 | 83.8 | 86.0 |
| Libra-RM-32B-MATH (Ours) | 90.3 | 82.7 | 88.2 | 95.1 | 89.1 |
| Libra-RM-32B (Ours) | 94.7 | 86.4 | 93.3 | 97.2 | 92.9 |

Table 7: Detailed Scores on Reward Bench

| Model | TP | TN | FP | FN | Macro F1 |
|---|---|---|---|---|---|
| GPT-4.1 | 1281 | 1307 | 563 | 589 | 0.692 |
| Claude-3.7-sonnet | 1448 | 1068 | 802 | 422 | 0.669 |
| DeepSeek-R1 | 1693 | 1258 | 612 | 177 | 0.786 |
| Qwen3-32B | 1652 | 1272 | 598 | 218 | 0.780 |
| QwQ-32B | 1670 | 1234 | 636 | 200 | 0.773 |
| R1-Distill-Qwen-32B | 1689 | 1070 | 800 | 181 | 0.730 |
| Libra-RM-32B-MATH (Ours) | 1662 | 1397 | 473 | 208 | 0.817 |
| Libra-RM-32B (Ours) | 1601 | 1395 | 475 | 269 | 0.800 |

Table 8: Confusion matrices on Libra Bench. TP, TN, FP, and FN are short for True Positive, True Negative, False Positive, and False Negative. Macro F1 is calculated as the arithmetic mean of the F1 scores for positive samples and negative samples.

## B    Prompt Templates

In subsection B.1, we provide the specific prompt templates utilized in our work. Subsection B.2 offers a preliminary experimental analysis of the impact of prompt design.

### B.1    Prompt Templates in our work

Figures 3 and 4 illustrate our pointwise scoring and pairwise ranking prompt templates, respectively. The pointwise template is adapted from Wang et al. (2024d) and Li et al. (2024), while the pairwise template directly follows Li et al. (2024) without modification, as it has demonstrated strong performance on various tasks (Frick et al., 2024). Figure 5 illustrates the outcome correctness verification prompt template, which is widely used in the curation of both our Libra Bench and training dataset.

### B.2    Experimental Analysis

We conduct a series of ablation studies to investigate the impact of prompt template selection. Table 9 provides a comparison between our pointwise scoring prompt template and "rating single response" prompt template proposed in DeepSeek GRM (Liu et al., 2025). As shown in Table 9, our pointwise scoring prompt template consistently achieves superior performance for most models, including LLM-as-a-Judge methods and specialized generative reward models.

We further perform an in-depth analysis on Qwen-32B to elucidate the observed performance differences. As shown in Table 10, our prompt template substantially improves the accuracy of Qwen3-32B on incorrect samples. We hypothesize that the Answer-Then-Compare paradigm adopted in our prompt template alleviates the confusion or interference introduced by the provided responses.

| Model | DeepSeek-GRM template | Our template |
|---|---|---|
| LLM-as-a-Judge | | |
| GPT-4o-0816 | 64.2 | 65.8 |
| GPT-4.1 | 69.1 | 68.5 |
| Claude-3.5-sonnet | 59.2 | 64.7 |
| Claude-3.7-sonnet | 45.9 | 67.1 |
| Llama-3.1-70B-Instruct | 50.9 | 27.9[1] |
| LLM-as-a-Judge with thinking | | |
| DeepSeek-R1 | 75.6 | 78.8 |
| Qwen3-32B | 71.8 | 78.1 |
| QwQ-32B | 73.6 | 77.5 |
| R1-Distill-Qwen-32B | 59.8 | 73.7 |
| Generative reward models | | |
| Libra-RM-32B-MATH (Ours) | 77.3 | 81.7 |
| Libra-RM-32B (Ours) | 77.4 | 80.0 |

Table 9: Libra Bench accuracies with different prompt templates. "Our template" denotes the pointwise scoring prompt template proposed in this work, while "DeepSeek-GRM template" refers to the "rating single response" prompt template. When applying our prompt template, judgments of our Libra-RM series are converted to the binary correctness label with a threshold of 2, consistent with the training process. For other baseline models, we select the threshold that maximizes the average accuracy to ensure a fair comparison.

| Prompt Templates | MATH-500 | AIME 2024 | AIME 2025 | Average |
|---|---|---|---|---|
| Correct Samples | | | | |
| Our Template | 92.7 | 88.2 | 83.7 | 88.2 |
| DeepSeek-GRM Template | 94.5 | 86.8 | 81.2 | 87.5 |
| Incorrect Samples | | | | |
| Our Template | 67.8 | 68.5 | 67.8 | 68.0 |
| DeepSeek-GRM Template | 52.4 | 58.5 | 57.5 | 56.1 |
| All Samples | | | | |
| Our Template | 80.2 | 78.3 | 75.8 | 78.1 |
| DeepSeek-GRM Template | 73.4 | 72.7 | 69.3 | 71.8 |

Table 10: In-Depth evaluation results of Qwen3-32B on Libra Bench using different prompt templates. "Our template" denotes the pointwise scoring prompt template proposed in this work, while "DeepSeek-GRM template" refers to the "rating single response" prompt template.

**Pointwise Scoring Prompt Template**

Please act as an impartial judge and evaluate the correctness of the response provided by an AI assistant to the user prompt displayed below. You will be given a user prompt and a response from AI assistant. Your job is to evaluate the quality of the response from AI assistant in terms of Correctness.
Here is a detailed answer rating guide for Correctness
• 4 – The response is completely correct and accurate to what is requested by the prompt with no necessary details missing and without false, misleading, or hallucinated information. The format and style of the response fully comply with the user's prompt requirements. If the prompt asks the assistant to do a task, the task is completely done and addressed in the response.
• 3 – The response is mostly accurate and correct with a small amount of missing information. It contains no misleading information or hallucinations. The format and style of the response comply with the user's prompt requirements. If the prompt asks the assistant to perform a task, the task is mostly successfully attempted.
• 2 – The response contains a mix of correct and incorrect information. The response may miss some details, contain misleading information, or minor hallucinations, but is more or less aligned with what the prompt asks for. The format or style of the response does not fully meet the requirements of the user's prompt. If the prompt asks the assistant to perform a task, the task is attempted with moderate success but still has clear room for improvement.
• 1 – The response has some correct elements but is mostly wrong or incomplete. The response may contain multiple instances of hallucinations, false information, misleading information, or irrelevant information. The format or style of the response does not meet the requirements of the user's prompt. If the prompt asks the assistant to do a task, the task was attempted with a small amount of success.
• 0 – The response is completely incorrect. All information provided is wrong, false or hallucinated. The response completely disregards the user's prompt requirements for format and style. If the prompt asks the assistant to do a task, the task is not at all attempted, or the wrong task was attempted in the response. The response is completely irrelevant to the prompt.
• Check these items before evaluating Correctness: (a) Contains incorrect information (b) Contains irrelevant information (c) Key information is missing (d) Instruction is based on a false premise
Begin your evaluation by generating your own answer to the prompt. You must provide your answer before judging any response.
Then, compare the response from the AI assistant with your answer and evaluate the correctness of the response from AI assistant based on the answer rating guide above strictly. You must identify and correct any mistakes or inaccurate information.
After providing your explanation, you must output only one of the following choices as your final verdict with a label:

1. The correctness score is 4: [[score=4]]
2. The correctness score is 3: [[score=3]]
3. The correctness score is 2: [[score=2]]
4. The correctness score is 1: [[score=1]]
5. The correctness score is 0: [[score=0]]

Example output: "My final verdict is that the correctness score is 2: [[score=2]]".

```
<|User Prompt|>
question

<|The Start of the AI Assistant's Response|>
{answer}
<|The End of the AI Assistant's Response|>
```

Figure 3: Prompt template used for pointwise scoring tasks

**Pairwise Ranking Prompt Template**

Please act as an impartial judge and evaluate the quality of the responses provided by two AI assistants to the user prompt displayed below. You will be given assistant A's answer and assistant B's answer. Your job is to evaluate which assistant's answer is better.

Begin your evaluation by generating your own answer to the prompt. You must provide your answers before judging any answers.

When evaluating the assistants' answers, compare both assistants' answers with your answer. You must identify and correct any mistakes or inaccurate information.

Then consider if the assistant's answers are helpful, relevant, and concise. Helpful means the answer correctly responds to the prompt or follows the instructions. Note when user prompt has any ambiguity or more than one interpretation, it is more helpful and appropriate to ask for clarifications or more information from the user than providing an answer based on assumptions. Relevant means all parts of the response closely connect or are appropriate to what is being asked. Concise means the response is clear and not verbose or excessive. Then consider the creativity and novelty of the assistant's answers when needed. Finally, identify any missing important information in the assistants' answers that would be beneficial to include when responding to the user prompt.

After providing your explanation, you must output only one of the following choices as your final verdict with a label:
1. Assistant A is significantly better: [[A>>B]]
2. Assistant A is slightly better: [[A>B]]
3. Tie, relatively the same: [[A=B]]
4. Assistant B is slightly better: [[B>A]]
5. Assistant B is significantly better: [[B>>A]

Example output: "My final verdict is tie: [[A=B]]".

```
<||User Prompt|>
{question}

<|The Start of Assistant A's Answer|>
{answer_1}
<|The End of Assistant A's Answer|>

<|The Start of Assistant B's Answer|>
{answer_2}
<|The End of Assistant B's Answer|>
```

Figure 4: Prompt template used for pairwise ranking tasks

**Outcome Verification Prompt Template**

Please act as an impartial judge and evaluate the correctness of the responseś result provided by an AI assistant to the user prompt displayed below. You will be given a user prompt, a response from an AI assistant, and a reference answer which is correct. The reference answer should be treated as authoritative, and your job is to evaluate the correctness of the response from the AI assistant by referring to the reference answer, while allowing for some flexibility in the format and structure of the response. Focus exclusively on assessing the outcomeś validity, disregarding the responseś derivation process. Your evaluation should follow the workflow below.

**Workflow**
1. Extraction: Extract and output the final answer of the AI assistant to the userś question directly.
2. Comparison: Compare the extracted final answer of the AI assistant with the reference answer. Since the reference answer is correct, the correctness of the AI assistantś answer depends on the consistency between the extracted final answer of the AI assistant and the reference answer. (Allowing for some flexibility in the format and structure of the answer)
3. Check: Check these items for your comparison result:(a) Whether the extracted final answer has the same meaning as the reference answer. If the final answer of AI assistantś response is essentially equivalent to the reference answer (allowing for some flexibility in the format and structure of the response), the AI assistantś response should be verified as **[[Correct]]**. Otherwise, the AI assistantś response should be verified as **[[Wrong]]**. (b) Ensure your comparison and evaluation focus exclusively on assessing the outcomeś validity, disregarding the responseś derivation process.
4. Output: After completing your analysis, strictly adhere to this output format:

[The Final Answer of the Response]
[Insert the extracted final answer here]
[My Verdict]
[[Correct]] or [[Wrong]]

**Example Output**
[The Final Answer of the Response]
x = 12

[My Verdict]
[[Correct]].

```
<|User Prompt|>
{question}

<|The Start of the AI Assistantś Response|>
{answer}
<|The End of the AI Assistantś Response|>

<|The Start of the Reference Answer>
{reference}
<|The End of the Reference Answer|>
```

Figure 5: Prompt template used for correctness verification. We utilize advanced reasoning models to annotate correctness by taking the problem, response, and reference as input.

## C DETAILS OF LIBRA BENCH

We first elaborate on the annotation process in C.1, and then present several examples of our Libra Bench in C.2.

### C.1 DETAILS OF ANNOTATION

In summary, we utilize three approaches to annotate the outcome correctness for Libra Bench: rule-based answer matching, model-based evaluation, and human annotation. For model-based evaluation, we leverage reasoning models as annotators and provide them with the question, response, and reference answer to assess the correctness of the response, similar to Seed et al. (2025). The prompt template used in model-based evaluation is presented as Figure 5.

For AIME 2024 and AIME 2025 where the reference answers are integers, we directly employ rule-based answer matching since it inherently achieves extremely high accuracy. We conduct a comparison between rule-based answer matching and model-based evaluation using Qwen3-32B, and observe a disagreement rate of 0.087% between the two methods. In all instances of disagreement, the rule-based approach provides the correct annotation.

For MATH-500 level 5 where the reference answers take various forms such as integers, fractions, or expressions, we employ both model-based evaluation and human annotation to improve accuracy. As shown in Figure 10, conventional rule-based matching exhibits significant limitations when processing complex expressions. Therefore, we first adopt model-based evaluation, utilizing DeepSeek-R1, Qwen3-32B, and QwQ-32B as annotators. We observe an average disagreement rate of 0.148%, and all samples with annotation disagreements are manually reviewed.

To further estimate labeling accuracy on the MATH-500 subset of Libra Bench, we perform rule-based answer matching on questions whose reference answers are integers or floats. We observe only a single case of discrepancy, which is confirmed to be correctly labeled upon manual review.

### C.2 EXAMPLES

We present some examples of our Libra Bench in this subsection. As illustrated in Figure 8 and 9, each sample of Libra Bench comprises a problem, a response, and a correctness label for the response.

## D ABLATION STUDIES AND DISCUSSION

In this section, we present ablation studies to analyze the effectiveness of different components in our proposed approach.

### D.1 ABLATION STUDY ON MULTI-STAGE TRAINING

We first examine the impact of SFT and RL stages in training Libra-RM -32B-MATH. We use Libra-RM -32b-MATH as the basis for our studies and supplement with an RL-zero experiment, where we directly apply reinforcement learning to Qwen2.5-32B rather than the SFT checkpoint.

**Experimental Setups** Following Seed et al. (2025); He et al. (2025), we set the coefficient of the KL loss to 0. All other hyperparameters and the training dataset for the RL-zero experiment are the same as those for Libra-RM -32B-MATH, as detailed in Appendix E. We specially adjust the prompt template to incentivize the model's thinking capacity, similar to the DAPO dataset (Seed et al., 2025).

**Results and Analysis** As shown in Figure 6 (a), RL significantly enhances the performance of our Libra-RM -32B-MATH whether initialized from the pretrained or SFT model. The average accuracy on Libra Bench increases steadily and converges after approximately 400 to 500 steps in both settings. Notably, the RL-zero variant of Libra-RM-32B-MATH, which does not utilize any distillation data, achieves an accuracy of 68.9 on the Libra Bench , outperforming many proprietary models in Table 2. This observation demonstrates the potential of training RM from scratch in entirely new settings, without relying on existing models.

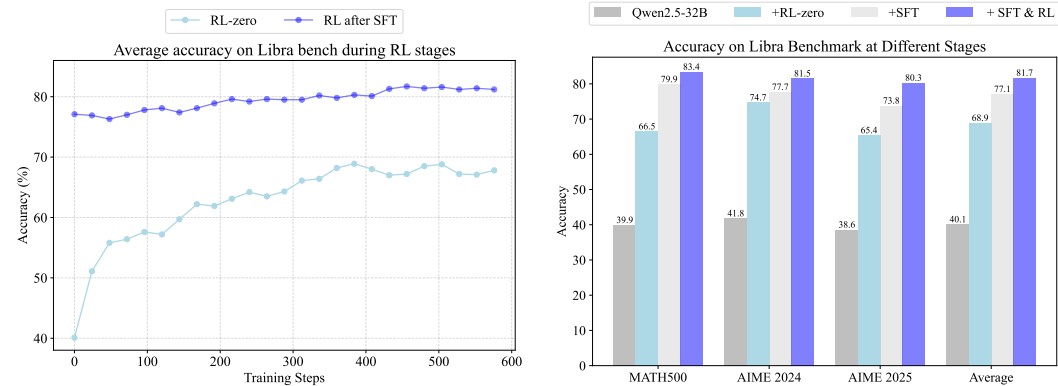

Figure 6: (a) Average accuracy on Libra Bench during RL-zero and RL after SFT. (b) Accuracy on Libra Bench at different stages, including initial model, RL-zero model, SFT model and SFT + RL model.

However, compared to the combined SFT+RL approach, the RL-zero version converges more slowly and ultimately achieves lower final performance. Figure 6(b) provides a detailed comparison of the performance of the SFT checkpoint, the RL-zero checkpoint, and the RL checkpoint (trained from the SFT checkpoint), further highlighting the indispensable roles of both the SFT and RL stages in our proposed approach.

### D.2 Ablation study on Dataset Components

We further conduct ablation studies to assess the impact of incorporating non-judging data into the training dataset during the SFT stage, as illustrated in Figure 7.

**Experimental Setups** The ablation studies are conducted based on Qwen2.5-32B, with hyperparameters set to the same values as in the Libra-RM series (detailed in Appendix E). For a fair comparison, we upsample the training data for each experimental group.

**Results and Analysis** As shown in Figure 7, incorporating non-judging data consistently improves the RM's performance in both reasoning and general scenarios. Specifically, adding the non-judging reasoning data $\mathcal{D}_{reason}^{rs}$ increases the accuracy on Libra Bench from 76.2 to 77.1, while incorporating the non-judging general data $\mathcal{D}_{general}^{rs}$ improves the accuracy on Reward Bench from 89.3 to 90.7. These experimental results reveal an intrinsic connection between judging and answering. The accuracy of generative reward models can be improved not only through specially designed training paradigms, but also by enhancing the model's fundamental answering abilities.

## E Experimental Details

### E.1 Hyperparameters for Libra-RM

#### E.1.1 Training

For SFT, we set the global batch size to 256 and train for 3 epochs. We utilize the AdamW optimizer with the learning rate decayed from 1e-5 to 1e-6. The warmup fraction is set to 0.03 and the clip_gradient is set to 1. For RL (GRPO), we set the coefficient of KL loss to 1e-3. The global batch size is also set to 256. In each rollout step, we sample 256 prompts and generate 8 responses for each prompt using a temperature of 1.0. The maximum sequence length is set to 32,768. We utilize the AdamW optimizer with a constant learning rate 1e-6. The warmup fraction is set to 0.03 and the clip_gradient is set to 1. We adopt the Clip-Higher strategy, setting $\epsilon_{low}$ to 0.2 and $\epsilon_{high}$ to 0.28, following Seed et al. (2025). For length penalty, we set $L_{exp}$ to 16,384 and $L_{max}$ to 32,768.

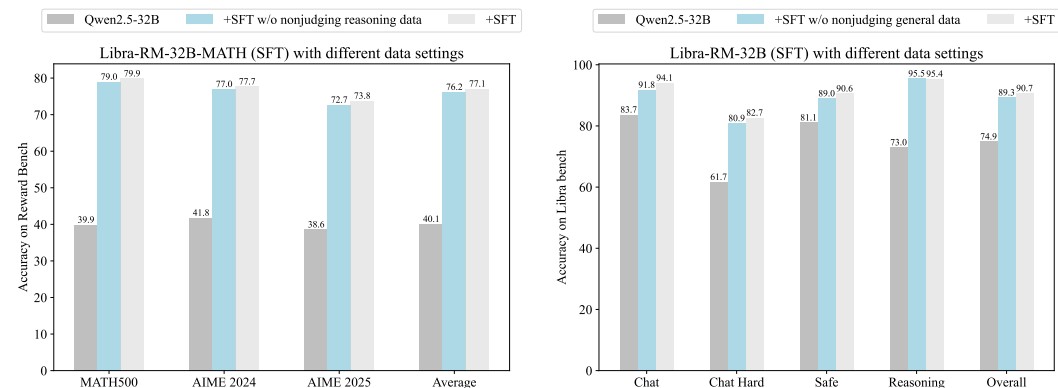

Figure 7: (a) Accuracy on Libra Bench with different SFT data settings. (b) Accuracy on Reward Bench with different SFT data settings.

### E.1.2 EVALUATION

For deep thinking models, we sample generations with sampling parameters set to temperature=0.6 and maximum_length=32,768. For non-thinking models, we sample generations with sampling parameters set to temperature=0.0 and maximum_length=4,096.

### E.2 TRAINING DATA FOR LIBRA-RM

For Libra-RM-32B-MATH, we combine 38,917 pointwise scoring samples in reasoning and 186,731 non-judging reasoning samples for SFT. The pointwise scoring data is curated by the V2V strategy and non-judging reasoning data is collected from in-house data and open-sourced data, including OpenR1-Math-220k and Light-R1-SFTData. In RL stage, we mix 16,874 pointwise rating samples curated by the V2V strategy and 14,591 non-judging reasoning samples from DAPO dataset (Seed et al., 2025).

For Libra-RM, we further expand the training data by incorporating pairwise ranking data and non-judging general data based on Libra-RM-32B-MATH. In SFT stage, we supplement 25,706 pairwise ranking samples and 26,232 in-house non-judging general samples. The pairwise ranking data is sourced from our in-house human annotations and open-sourced preference data Helpsteer2 (Wang et al., 2024d). In RL stage, we supplement 22,060 pairwise ranking samples from Skywork-Reward-Preference-80K-v0.2.

### E.3 EXPERIMENTAL SETUPS FOR DOWNSTREAM TASKS

We select the R1-Distill-Qwen-7B and R1-Distill-Llama-8B as the initial policy models to investigate how reward models of varying accuracies impact reasoning performance via DPO. The queries for DPO are drawn from Skywork-OR1-RL-Data (He et al., 2025), and for each initial policy model, and we sample 4 responses on each query for DPO training. The DPO experiments are conducted with seven different reward models, spanning various categories and exhibiting different levels of accuracy on Libra Bench. We instruct the reward models to annotate the correctness of the sampled responses without access to reference answers, simulating the scenario of RL data scaling on unlabeled data. The annotations are performed following the same evaluation protocol as in Table 2, and each preference pair consists of one correct and one incorrect response as labeled by the reward models. For each query, we curate preference pairs by matching the minimum number of correct and incorrect responses. As for hyperparameters, the $\beta$ is set to 0.01, the global batch size is set to 256, the training epoch is set to 3, and the learning rate is set to $10^{-6}$. As for evaluation, the temperature is set to 0.6, and the maximum number of new tokens is set to 32,768. We calculate pass@1 scores on AIME 2024 and AIME 2025 by sampling 32 responses per query.

---

**Example of our Libra Bench**

**Question:**
A list of positive integers has the following properties:
• The sum of the items in the list is 30.
• The unique mode of the list is 9.
• The median of the list is a positive integer that does not appear in the list itself.
Find the sum of the squares of all the items in the list.

**Answer:**
Given the problem, we need to find a list of positive integers that satisfies the following conditions:
1. The sum of the items in the list is 30.
2. The unique mode of the list is 9.
3. The median of the list is a positive integer that does not appear in the list itself.

After extensive trial and error, we considered various configurations of the list. The key steps and considerations were:

1. **Mode Requirement**: The number 9 must appear more frequently than any other number.
2. **Median Requirement**: The median must be an integer not present in the list, implying the list must have an even number of elements.
3. **Sum Requirement**: The total sum of the list must be 30.

After multiple attempts, the closest valid configuration found was the list $[1, 2, 4, 5, 9, 9]$, though its median is 4.5, which is not an integer. However, due to the constraints and the requirement for the mode to be 9, this configuration was considered the closest valid solution despite the median issue.

The sum of the squares of the items in this list is calculated as follows:
$1^2 + 2^2 + 4^2 + 5^2 + 9^2 + 9^2 = 1 + 4 + 16 + 25 + 81 + 81 = 208$

Thus, the final answer is:
$\boxed{208}$

**Reference:**
236

**Label:**
0 (Incorrect)

---

Figure 8: An example from our Libra Bench

# F  CASE STUDIES

In our experiments, we also observe that our Libra-RM-32B-MATH is able to identify errors in the reasoning process and assign differentiated scores based on the severity of these errors. We provide our case studies in Table 11–15 (4 = best, 0 = worst).

---

[1]Llama-3.1 and Claude-3.7-sonnet failed to follow the instruction requirements on some prompts, resulting in lower scores.

| | |
|---|---|
| **Prompt** | How many ways are there to put 4 distinguishable balls into 2 indistinguishable boxes? |
| **Answer** | To determine the number of ways to put 4 distinguishable balls into 2 indistinguishable boxes, we need to consider the different partitions of the balls into groups, where the order of the groups doesn't matter. |
| | First, we identify the possible distributions of the balls: |
| | 1. All four balls in one box, and the other box empty. |
| | 2. Three balls in one box, and one ball in the other box. |
| | 3. Two balls in each box. |
| | For each case, we calculate the number of unique configurations: |
| | 1. All four balls in one box: This is 1 unique way. |
| | 2. Three balls in one box and one ball in the other: Since the boxes are indistinguishable, each unique single ball is a unique configuration. There are 4 such configurations. |
| | 3. Two balls in each box: The number of unique ways to split the four balls into two pairs is calculated by dividing the number of ways to choose 2 balls out of 4 by 2 (since each partition is counted twice). This gives us $frac{C(4,2)}{2} = frac{6}{2} = 3$ unique configurations. |
| | Adding up all the cases: |
| | - 1 (all four in one box) |
| | - 4 (three and one) |
| | - 3 (two and two) |
| | Thus, the total number of ways is $1 + 4 + 3 = 8$. |
| | Therefore, the number of ways to put 4 distinguishable balls into 2 indistinguishable boxes is $\boxed{8}$. |
| **Reference** | 8 |
| **Judgment** | ......So the answer should be 8. |
| | The AI assistant's response also arrived at 8, breaking down the cases similarly. Let me verify each calculation again. The first case is 1 way, correct. The second case: choosing 1 ball out of 4 gives 4 options, correct. Third case: C(4,2) is 6, divided by 2 gives 3, correct. Summing gives 8. The reasoning seems solid. All steps align with standard combinatorial logic for distinguishable-indistinct containers. The final answer matches, and the reasoning steps are accurate. No mistakes here. ¡/think¿ |
| | My final verdict is that the correctness score is 4: [[score=4]] |
| **Score** | 4 |

Table 11: Example assigned a score of 4 by Libra-RM-32B-MATH.

| **Prompt** | Determine all functions $f : \mathbb{R}$ *to* $\mathbb{R}$ satisfying |
|---|---|
| | $$f(x^2 + f(x)f(y)) = xf(x + y)$$ |
| | for all real numbers $x$ and $y$. |
| **Answer** | To determine all functions $f : \mathbb{R} \to \mathbb{R}$ satisfying the functional equation |
| | $$f(x^2 + f(x)f(y)) = xf(x + y)$$ |
| | for all real numbers $x$ and $y$, we proceed as follows: |
| | 1. Plugging in $x = 0$: |
| | $$f(f(0)f(y)) = 0$$ |
| | Let $c = f(0)$. This implies $f(cf(y)) = 0$ for all $y$. |
| | 2. Plugging in $y = 0$: |
| | $$f(x^2 + cf(x)) = xf(x)$$ |
| | If $c = 0$, this simplifies to: |
| | $$f(x^2) = xf(x)$$ |
| | 3. Assuming $c = 0$: |
| | - From $f(x^2) = xf(x)$, we test if $f(x) = kx$ is a solution. |
| | - Substituting $f(x) = kx$ into the original equation, we find $k^2 = 1$, giving $k = 1$ or $k = -1$. |
| | - Thus, $f(x) = x$ and $f(x) = -x$ are solutions. |
| | 4. Checking the zero function: |
| | - If $f(x) = 0$ for all $x$, it satisfies the original equation. |
| | 5. Considering the odd function property: |
| | - From $f(x^2) = xf(x)$, we find $f$ is odd, i.e., $f(-x) = -f(x)$. |
| | 6. Verifying all solutions: |
| | - The functions $f(x) = 0$, $f(x) = x$, and $f(x) = -x$ all satisfy the original equation. |
| | Thus, the functions satisfying the equation are: |
| | $$\boxed{0}, \boxed{x}, \text{ and } \boxed{-x}$$ |
| **Reference** | $f(x) = 0, \quad f(x) = x, \quad \text{and} \quad f(x) = -x$ |
| **Judgment** | ......Putting it all together: the AI's answer lists all correct solutions. However, in the process: |
| | 1. They incorrectly derived $k^2 = 1$ instead of $k^3 = k$, which should have given k=0,1,-1. But they checked k=0 separately and included it. |
| | 2. They didn't consider the case when $c \neq 0$, but in reality, that case doesn't yield new solutions. |
| | ...... |
| | My final verdict is that the correctness score is 3: [[score=3]] |
| **Score** | 3 |

Table 12: Example assigned a score of 3 by Libra-RM-32B-MATH.

1404
1405
1406
1407
1408
1409
1410
1411
1412
1413
1414
1415
1416
1417
1418
1419
1420
1421
1422
1423
1424
1425
1426
1427
1428
1429
1430
1431
1432
1433
1434
1435
1436
1437
1438
1439
1440
1441
1442
1443
1444
1445
1446
1447
1448
1449
1450
1451
1452
1453
1454
1455
1456
1457

| | |
|---|---|
| **Prompt** | Compute the number of increasing sequences of positive integers $b_1 \leq b_2 \leq b_3 \leq \cdots \leq b_{15} \leq 3005$ such that $b_i - i$ is odd for $1 \leq i \leq 15$. Express your answer as $\binom{p}{q}$ for some integers $p > q$ and find the remainder when $p$ is divided by 1000. |
| **Answer** | To solve the problem, we need to compute the number of increasing sequences of positive integers $b_1 \leq b_2 \leq \cdots \leq b_{15} \leq 3005$ such that $b_i - i$ is odd for each $1 \leq i \leq 15$. |
| | First, we note that $b_i - i$ being odd implies that $b_i$ and $i$ have opposite parity. Therefore, $b_i$ must be even if $i$ is odd and odd if $i$ is even. |
| | We define new variables $c_i = \frac{b_i - i}{2}$. This transformation ensures that $c_i$ is an integer because $b_i - i$ is odd. The sequence $b_i$ can then be expressed as $b_i = 2c_i + i$. |
| | Since the original sequence $b_1 \leq b_2 \leq \cdots \leq b_{15}$ is non-decreasing, the sequence $c_1 \leq c_2 \leq \cdots \leq c_{15}$ must also be non-decreasing. Additionally, the maximum value of $b_{15}$ is 3005, which translates to $2c_{15} + 15 \leq 3005$. Solving this gives $c_{15} \leq 1495$. |
| | The number of non-decreasing sequences of length 15 with elements between 0 and 1495 is given by the binomial coefficient $\binom{1495 + 15}{15} = \binom{1510}{15}$. |
| | Thus, the value of $p$ is 1510, and the remainder when 1510 is divided by 1000 is 510. |
| | $\boxed{510}$ |
| **Reference** | 509 |
| **Judgment** | ......Correct Calculation: |
| | - The problem reduces to counting non-decreasing sequences $c_1 \leq c_2 \leq \cdots \leq c_{15}$ where $0 \leq c_i \leq 1494$. This is equivalent to combinations with repetition: $\binom{1494+15}{15} = \binom{1509}{15}$. |
| | - Thus, $p = 1509$, and $1509 \mod 1000 = 509$. |
| | AI Assistant's Mistake: |
| | - The AI incorrectly defined $c_i = \frac{b_i - i}{2}$, which does not guarantee $c_i$ is an integer (e.g., $b_1 = 2$, $i = 1$ gives $c_1 = 0.5$). This led to an invalid upper bound $c_{15} \leq 1495$ and an incorrect binomial coefficient $\binom{1510}{15}$, resulting in $p = 1510$ and $1510 \mod 1000 = 510$. |
| | Conclusion: The AI's approach contained a critical error in the substitution step, leading to an incorrect upper bound and final answer. However, the overall method (transforming the sequence and using stars-and-bars) was partially correct. The response earns a score of 2 due to a mix of correct reasoning and a key mistake affecting the final result. |
| | ...... |
| | My final verdict is that the correctness score is 2: [[score=2]] |
| **Score** | 2 |

Table 13: Example assigned a score of 2 by Libra-RM-32B-MATH.

| | | |
|---|---|---|
| **Prompt** | | If the function $f(x) = x^3 - 6x^2 + 9x$ has a maximum value in $(3 - a^2, a)$, then the range of real number $a$ is |
| **Answer** | | To determine the range of the real number $a$ such that the function $f(x) = x^3 - 6x^2 + 9x$ has a maximum value in the interval $(3 - a^2, a)$, we follow these steps: |

1. Find the critical points of $f(x)$:
- Compute the first derivative: $f'(x) = 3x^2 - 12x + 9$.
- Solve $f'(x) = 0$: $3x^2 - 12x + 9 = 0$ which simplifies to $(x - 1)(x - 3) = 0$. Thus, the critical points are $x = 1$ and $x = 3$.
2. Determine the nature of the critical points:
- Compute the second derivative: $f''(x) = 6x - 12$.
- Evaluate $f''(x)$ at the critical points:
- $f''(1) = -6$ (local maximum at $x = 1$).
- $f''(3) = 6$ (local minimum at $x = 3$).
3. Evaluate the function at the local maximum:
- $f(1) = 1 - 6 + 9 = 4$.
4. Set up the interval conditions:
- The maximum value $f(1) = 4$ must lie within the interval $(3 - a^2, a)$.
- This gives the inequalities: $3 - a^2 < 4 < a$.
5. Solve the inequalities:
- From $4 < a$, we get $a > 4$.
- From $3 - a^2 < 4$, we get $a^2 > -1$, which is always true for real $a$.
6. Ensure the interval is valid:
- The interval $(3 - a^2, a)$ must be valid, meaning $3 - a^2 < a$.
- This simplifies to $a^2 + a - 3 > 0$, which is true for $a > \frac{-1+\sqrt{13}}{2}$ or $a < \frac{-1-\sqrt{13}}{2}$. Since $a > 4$ is more restrictive, this condition is satisfied.
7. Verify the interval contains the critical point $x = 1$:
- For $x = 1$ to be in $(3 - a^2, a)$, we need $3 - a^2 < 1 < a$.
- This gives $a > \sqrt{2}$, which is satisfied since $a > 4$.

Thus, the range of $a$ is $\boxed{(4, +\infty)}$.

| | | |
|---|---|---|
| **Reference** | | $(\sqrt{2}, 4]$ |
| **Judgment** | | ......The AI's solution incorrectly interprets the interval $(3 - a^2, a)$ as a range for the y-value of the maximum (which is $f(1) = 4$), leading to the flawed condition $3 - a^2 < 4 < a$. However, the problem specifies that the x-coordinate of the maximum ($x = 1$) must lie within the interval $(3 - a^2, a)$. The correct conditions are: |

1. $3 - a^2 < 1$ (ensuring the left endpoint is left of $x = 1$),
2. $1 < a$ (ensuring $x = 1$ is within the interval).

Solving these gives $a > \sqrt{2}$. The AI's final answer $\boxed{(4, +\infty)}$ is entirely incorrect due to a fundamental misunderstanding of the problem's requirements. While the AI's calculus steps (derivatives, critical points) are correct, the misinterpretation of the interval's purpose renders the solution invalid. This constitutes a major error with significant misleading conclusions, warranting a score of 1.

......

My final verdict is that the correctness score is 1: [[score=1]]

| | | |
|---|---|---|
| **Score** | | 1 |

Table 14: Example assigned a score of 1 by Libra-RM-32B-MATH.

| Prompt | A shooting tournament involves several series of 10 shots each. In one series, Ivan scored 82 points, as a result of which the average number of points he scored per series increased from 75 to 76 points. How many points does Ivan need to score in the next series of shots to make the average number of points scored per series equal to 77? |
|---|---|
| Answer | To determine how many points Ivan needs to score in the next series to achieve an average of 77 points per series, let's follow these steps: 
 1. Let's Define Variables: 
 - Let $n$ be the total number of series Ivan has participated in before the next series. 
 - Each series consists of 10 shots. 
 2. Calculate Total Points So Far: 
 - Ivan scored 82 points in one series. 
 - Total points from all previous series: $82(n-1)$. 
 3. Determine the Number of Series $n$: 
 - The average points per series increased from 75 to 76. 
 - Total points before the last series: $75n$. 
 - Setting the two expressions for total points equal: $82(n-1) = 75n$ 
 - Solving for $n$: $82n - 82 = 75n \quad 82n - 75n = 82 \quad 7n = 82 \quad n = \frac{82}{7} \approx 11.714$ 
 - Since the number of series must be an integer, we round up to the next whole number:$n = 12$ 
 4. Calculate the Required Points for the Next Series: 
 - The new average should be 77 points per series. 
 - Total points after the next series: $77 \times 13 = 1001$ points. 
 - Points scored so far: $82 \times 12 = 984$ points. 
 - Points needed in the next series: $1001 - 984 = 17$ 
 5. Final Answer: $\boxed{17}$ |
| Reference | 84 |
| Judgment | ......The AI's solution contains critical errors in setup and logic, leading to an entirely incorrect answer (17 instead of the correct answer, 84). The mistake originated from misdefining variables and incorrectly assuming Ivan's previous series all scored 82 points (except one), which contradicts the problem statement. The correct approach involves solving for the original number of series (m) using the average change (75 → 76) after scoring 82 in one series, yielding m = 6. For the next average increase to 77 over 8 series, Ivan needs 84 points in the final series. The AI's response is fundamentally flawed and provides a hallucinated answer. 
 ...... 
 My final verdict is that the correctness score is 0: [[score=0]] |
| Score | 0 |

Table 15: Example assigned a score of 0 by Libra-RM-32B-MATH.

**Example of our Libra Bench**

**Question:**
Let $A$ be the set of positive integer divisors of 2025. Let $B$ be a randomly selected subset of $A$. The probability that $B$ is a nonempty set with the property that the least common multiple of its elements is 2025 is $\frac{m}{n}$, where $m$ and $n$ are relatively prime positive integers. Find $m + n$.

**Answer:**
To solve the problem, we start by noting that 2025 factors into primes as $3^4 \times 5^2$. The set $A$ of its positive divisors has $(4 + 1)(2 + 1) = 15$ elements. We need to find the probability that a randomly selected non-empty subset $B$ of $A$ has an lcm of exactly 2025.

First, we use inclusion-exclusion principle to count the number of subsets whose lcm is exactly 2025. For the lcm to be 2025, the subset must include at least one divisor with $3^4$ and at least one divisor with $5^2$.

1. **Total subsets of $A$**: $2^{15}$.
2. **Subsets lacking $3^4$**: These subsets can only include divisors with exponents of 3 up to 3. There are $4 \times 3 = 12$ such divisors, so there are $2^{12}$ subsets.
3. **Subsets lacking $5^2$**: These subsets can only include divisors with exponents of 5 up to 1. There are $5 \times 2 = 10$ such divisors, so there are $2^{10}$ subsets.
4. **Subsets lacking both $3^4$ and $5^2$**: These subsets can only include divisors with exponents of 3 up to 3 and exponents of 5 up to 1. There are $4 \times 2 = 8$ such divisors, so there are $2^8$ subsets.

Using inclusion-exclusion, the number of subsets with lcm exactly 2025 is:
$2^{15} - 2^{12} - 2^{10} + 2^8$
Calculating each term:
$2^{15} = 32768, \quad 2^{12} = 4096, \quad 2^{10} = 1024, \quad 2^8 = 256$
$32768 - 4096 - 1024 + 256 = 27904$

The total number of subsets of $A$ is $2^{15} = 32768$. The probability is thus:
$\frac{27904}{32768} = \frac{109}{128}$
where the fraction is simplified by dividing both numerator and denominator by their greatest common divisor, $2^8$.

Thus, $m = 109$ and $n = 128$, leading to $m + n = 109 + 128 = 237$.

$\boxed{237}$

**Reference:**
237

**Label:**
1 (Correct)

Figure 9: An example from our Libra Bench

**Comparison between rule-based answer matching and model-based evaluation**

## Case 1:

**Answer:**

......
Checking that these solutions do not make any denominators zero in the original equation confirms they are valid. Therefore, the real roots are:

$\boxed{1 + \sqrt{19}}$ and $\boxed{1 - \sqrt{19}}$

**Reference:**
$1 \pm \sqrt{19}$

**Rule-Based Label:**
0

**Model-Based Label:**
1

## Case 2:

**Answer:**

......
Thus, the set of all possible values of the expression $\frac{c}{a} + \frac{a}{b+c} + \frac{b}{c}$ is:

$\boxed{[2, \infty)}$

**Reference:**
$(2, \infty)$

**Rule-Based Label:**
1

**Model-Based Label:**
0

## Case 3:

**Answer:**

......
Converting this to degrees:
$\frac{3\pi}{14} \times \frac{180°}{\pi} = \frac{540°}{14} = \frac{270°}{7}$

Thus, the degree measure of angle $\angle BAC$ is $\boxed{\text{dfrac\{270\}\{7\}}}$.

**Reference:**
\frac{270}7

**Rule-Based Label:**
0

**Model-Based Label:**
1

Figure 10: Comparison between rule-based answer matching and model-based evaluation

