# OpenReview forum: "Libra: Assessing and Improving Reward Model by Learning to Think"
_ICLR.cc/2026/Conference — Submitted to ICLR 2026_

### Official Review · Reviewer_V8zg · 2025-10-25

**Soundness:** 4
**Presentation:** 3
**Contribution:** 2
**Rating:** 4
**Confidence:** 4

**Summary:**

This paper introduces a framework for evaluating and improving generative reward models (GRMs) in reasoning tasks. The framework comprises two main components: 1) a reward benchmark curated from outputs of SoTA reasoning models; and 2) a family of GRMs trained using a combination of SFT, rejected sampling, and RLVR. The authors demonstrate that Libra-RM achieves good performance on mainstream reward benchmarks and yields consistent gains on downstream tasks.

**Strengths:**

- The manuscript is well-written and easy to follow. The paper clearly details training data construction, RL reward formulation, and ablation setups.
- A reliable, reasoning-oriented reward model and benchmark would be highly valuable for advancing RL research in LLM reasoning.

**Weaknesses:**

- While the overall framework is well executed, the technical components—such as rejection sampling, GRPO-based RL, and SFT—are well-established in prior work. The contribution lies more in system integration and evaluation than in algorithmic innovation.
- The study’s downstream evaluation primarily focuses on advancing mathematical reasoning with DPO. This makes the claimed generality toward “unverifiable reasoning tasks” somewhat speculative. Additional evidence with online RL and domains where correctness is ambiguous would strengthen the claims.

**Questions:**

See weakness above.

---

### Official Review · Reviewer_mokn · 2025-10-27

**Soundness:** 2
**Presentation:** 2
**Contribution:** 2
**Rating:** 4
**Confidence:** 4

**Summary:**

This work introduces Libra Bench, a corpus designed for reward model learning and evaluation in mathematical reasoning tasks. To construct the dataset, the authors first select questions from existing sources such as AIME 2024 and AIME 2025. Five models (DeepSeek and Qwen variants) then generate responses to these questions, which are subsequently verified through rule-based, model-based, and manual validation methods. The benchmark is used to evaluate various reward modelling approaches, such as discriminative reward models and LLM-as-a-Judge, demonstrating that Libra Bench poses a substantial challenge. Furthermore, the proposed reward modelling method outperforms baseline approaches, and downstream evaluations show that a reward model trained with this method can effectively enhance policy post-training.

**Strengths:**

* Comprehensive baseline methods are presented in the experiments, including evaluating 15 different reward models on the proposed dataset and 6 different reward models for downstream applications.
* More than 200 questions with responses from 5 different DeepSeek and Qwen variants are collected and annotated in the corpus.

**Weaknesses:**

* The experimental results in Table 3 appear incomplete. A statistical significance test is missing, which is crucial for demonstrating that the reported improvements are meaningful. Some performance gains, such as those between Libra-RM-32B-MATH, DeepSeek-R1, and Qwen3-32B in Table 2, are relatively marginal. Without significance testing, it is difficult to assess the contribution of the proposed method.
* The experimental setup for the unverifiable reasoning scenario (Section 6) is insufficiently described. Based on Figure 2, the experiments seem to be conducted on AIME24/25, which are typically verifiable tasks.
* An analysis of the impact of different model responses used during training would be highly informative. Although the authors collect responses with a wide range of performance levels, the effect of this diversity remains unclear. For instance, is it more beneficial to train exclusively on responses from stronger models, or does incorporating error-prone responses also provide useful learning signals for reward modelling? Additionally, the diversity of models used for response collection could be improved; currently, only five LLMs from two variants are employed. Incorporating models from different families (e.g., Phi-2, Gemma, or Llama) would enhance dataset diversity and provide deeper insights into the generalizability of the proposed approach.

**Questions:**

* Typo: The reference prefixes of *4* and *E* in L403 are missing.
* Typo: A space should be removed after "Libra Bench" in L412.
* Typo: A missing space should be inserted before "As shown" in L421.
* Is the $D^{rl}_{rank}$ used by any reward model (mentioned in Section 4.3)?
* What will the len_penalty be when $L_{max}$ equal to $L_{exp}$ (equation 4)?

---

### Official Review · Reviewer_8a9q · 2025-10-30

**Soundness:** 4
**Presentation:** 4
**Contribution:** 3
**Rating:** 6
**Confidence:** 4

**Summary:**

This paper presents Libra, a framework for assessing and improving reward models in complex reasoning tasks. It introduces Libra Bench, a reasoning-oriented benchmark built from challenging math problems and advanced reasoning models, and Libra-RM, a generative reward model trained via a “learning-to-think” approach combining rejection sampling and reinforcement learning. Libra-RM achieves state-of-the-art performance across multiple RM benchmarks and shows strong correlation between Libra Bench accuracy and downstream RL improvements.

**Strengths:**

1. Proposes a clear and comprehensive framework integrating a new reasoning-oriented benchmark (Libra Bench) and generative reward models (Libra-RM). These benchmark is well-designed with challenging math problems and advanced reasoning models, effectively testing RM correctness.
2. The method is well-motivated and clearly presented. Experimental settings are thorough, showing strong state-of-the-art results and meaningful correlation with downstream reasoning performance.

**Weaknesses:**

1. The work does not appear highly original to me, as similar directions have been explored in prior studies such as [1] and [2]. It would be helpful if the authors could discuss these related works in more detail or include a comparison in the experiments.
2. The training data is mainly distilled from several reasoning models, so the generalization to unseen models or broader domains remains uncertain. Providing experiments or analysis on cross-model transferability would strengthen the paper.

[1] GRAM: A Generative Foundation Reward Model for Reward Generalization
[2] RM-R1: Reward Modeling as Reasoning

**Questions:**

I’m wondering whether the authors have explored how the ratio between judging and non-judging data affects training. Does varying this proportion influence the model’s stability or final accuracy?

---

### Official Review · Reviewer_77LG · 2025-10-31

**Soundness:** 1
**Presentation:** 3
**Contribution:** 2
**Rating:** 2
**Confidence:** 4

**Summary:**

Libra introduces a reasoning-oriented RM benchmark (Libra Bench) for evaluating Reward Models on reasoning tasks with rollouts sampled with the latest reasoning models. The authors also trained 2 generative thinking reward models which they demonstrate to be superior to competing approaches on the Libra Bench. Finally the effectiveness of the Libra-RM is demonstrated by training SLMs on preference pairs graded by different RMs.

**Strengths:**

The paper is relatively well motivated to investigate how reasoning-style generative RMs may be better judges on reasoning tasks than existing RM or LLM-as-judge.

The reported performance gains of the RMs on Libra Bench is very significant, indicating the effectiveness of the proposed method.

**Weaknesses:**

The biggest and potentially fatal issue of the current work is data contamination. In particular, both the benchmark (Libra Bench) and the RM (Libra RM) rely on rollouts generated by models in the R1 model facility (DeepSeek-R1, R1-distilled SLMs). As such, it is very difficult to gauge the degree to which the performance gain that Libra RM has over competing method is coming from data distribution or not.

Additionally, the various choices of Libra-RM experimentations are not ablated. For example, the inclusion of non-judging data and length penalty are both not ablated. As a result it is unclear whether any of such design choices are necessary or specific to the specific model type that Libra-RM is working with (thinking/reasoning generative reward models).

Perhaps a meta question, since these tasks in Libra Bench are verifiable (e.g. AIME), why is a reward model needed? To answer this question will require running RL experiment using Libra-RM vs. verifiable reward.

**Questions:**

For RM benchmarks, I highly encourage a human-in-the-loop approach where the ground truth answers are sampled from human responses rather than model responses to avoid potential issue of data contamination. In the case where data contamination is unavoidable, it is then critical to cleanly separate in distribution performance evaluation from out of distribution evaluations.

---

### Meta-Review · Area_Chair_xQjA · 2026-01-12

**Summary:**

Across the four reviews, the main decision-driving concerns are:

1. Potentially fatal contamination / in-distribution coupling.
R-77LG argues that both Libra Bench and Libra-RM depend on rollouts from the same “R1 facility” (DSPK-R1 and R1-distilled models), making it difficult to determine whether reported gains reflect genuine reward-model improvement versus distributional advantage.
2. Insufficient evidence for the paper’s  general claims.
Multiple reviewers (R-mokn, R-V8zg) indicate that downstream evaluation is primarily math + DPO, which weakly supports claims about unverifiable reasoning or more general domains where correctness is ambiguous. R-8a9q similarly questions generalization to unseen models or domains due to distillation-heavy training data.
3. Missing or incomplete ablations / significance .
R-77LG requests ablations for design choices (non-judging data, length penalty). R-mokn highlights missing statistical significance testing and incomplete reporting in at least one table, which affects confidence in “marginal” gains.

Given these, the primary concern is the contamination/ID-vs-OOD ambiguity, with secondary concern around scope/generalization evidence.

**Reviewer Concerns:**

Since the authors did not submit a rebuttal, none of the reviewers’ concerns can be considered “addressed by the rebuttal.”

**Reviewer Scores:**

- 77LG: 2 → 2 (no change).
- 8a9q: 6 → 5 (down slightly), in discussion, the contamination/generalization concern raised by 77LG/mokn would likely reduce confidence in “above threshold,” especially if no OOD separation is demonstrated.
- mokn: 4 → 4 (no change).
- V8zg: 4 → 4 (no change)

---

### Decision · Program_Chairs · 2026-01-26

Reject